

# Fear of eyes: triadic relation among social anxiety, trypophobia, and discomfort for eye cluster

Kengo Chaya[1], Yuting Xue[1], Yusuke Uto[1], Qirui Yao[1] and Yuki Yamada[2]

[1] Graduate School of Human-Environment Studies, Kyushu University, Fukuoka, Japan
[2] Faculty of Arts and Science, Kyushu University, Fukuoka, Japan

## ABSTRACT

Imagine you are being gazed at by multiple individuals simultaneously. Is the provoked anxiety a learned social-specific response or related to a pathological disorder known as trypophobia? A previous study revealed that spectral properties of images induced aversive reactions in observers with trypophobia. However, it is not clear whether individual differences such as social anxiety traits are related to the discomfort associated with trypophobic images. To investigate this issue, we conducted two experiments with social anxiety and trypophobia and images of eyes and faces. In Experiment 1, participants completed a social anxiety scale and trypophobia questionnaire before evaluation of the discomfort experienced upon exposure to pictures of eye. The results showed that social anxiety had a significant indirect effect on the discomfort associated with the eye clusters, and that the effect was mediated by trypophobia. Experiment 2 replicated Experiment 1 using images of human face. The results showed that, as in Experiment 1, a significant mediation effect of trypophobia was obtained, although the relationship between social anxiety and the discomfort rating was stronger than in Experiment 1. Our findings suggest that both social anxiety and trypophobia contribute to the induction of discomfort when one is gazed at by many people.

## INTRODUCTION

Trypophobia is an irrational fear of holes. Typically, trypophobic images are composed of holes such as the seed head of the lotus flower or a honeycomb. According to the trypophobia website (www.trypophobia.com), the word "trypophobic" is relatively new. Therefore, a well-accepted definition has yet to be given.

Trypophobic images, as currently understood, usually present hole-like patterns. However, in one pioneering study of trypophobia conducted by *Cole & Wilkins (2013)*, it was found that the images that give rise to aversive reactions are not always holes. They explained this phenomenon in terms of the "visual system as a spatial frequency analyzer" (*Maffei & Fiorentini, 1973*) based on the fact that trypophobic images contain relatively high contrast at midrange spatial frequencies. Although this spectral feature is not a sufficient condition for eliciting trypophobic aversion, as indicated by Cole and Wilkins, they found that even normal individuals are more sensitive and averse to trypophobic images than

Corresponding author
Kengo Chaya,
k.c.1992.4.4.3@gmail.com

normal images. That is, trypophobia is not limited to being a pathological phenomenon but may be a normal visual preferential tendency. As is well known, spatial frequencies influence perception in many domains such as visual illusions (*Giora & Gori, 2010*) or esthetic pleasure (*Vannucci, Gori & Kojima, 2014*). For example, the Ouchi illusion (*Ouchi, 1977*) is clearly related to the spatial frequency of stimuli (*Ashida, 2002*) and fixational eye movements (*Rucci et al., 2007*), which are necessary for avoiding perceptual fading (e.g., *Martinez-Conde et al., 2006*; *Martinez-Conde, Otero-Millan & Macknik, 2013*; *Costela et al., 2013*; *McCamy, Macknik & Martinez-Conde, 2014*).

Furthermore, *Cole & Wilkins (2013)* showed that the spectral properties shared by images of highly poisonous animals (such as the box jellyfish and the King cobra snake) are similar to those of trypophobic images. The finding supported the hypothesis that humans have developed the ability to detect stimuli based on an early, fast-reacting visual mechanism and that human survival has heavily relied on this mechanism to avoid potential threats from the surrounding environment throughout the history of human evolution. *Le, Cole & Wilkins (2015)* developed the Trypophobia Questionnaire (TQ). Thanks to the TQ, researchers have been able to investigate the individual differences in the trypophobic trait. Le, Cole, and Wilkins also showed that convex objects induce aversion as effectively as concave objects. This result supports the above investigation in that the spectral properties possessed by images are important in inducing trypophobic reactions. However, the above studies focused on only the perceptual aspect of trypophobia, but the underlying cognitive mechanism has not been revealed: Does the aversion to trypophobic images depend on the other personal traits of observers?

In the present study, we focused on social anxiety. Social anxiety disorder is the fear of a social situation that may involve negative judgment from others. Social anxiety disorder is a relatively common psychiatric disorder with a lifetime prevalence of 10% to 15% (*Kessler et al., 2005*). Previous studies revealed that people with social anxiety disorder have a fear of eye contact or being gazed at. For instance, the gaze cone for patients with social anxiety disorder is enlarged in the presence of other observers (*Gamer et al., 2011*). The more socially anxious a person is, the more frequently he or she feels that he or she is being looked at by others and reports reactive behavior, such as fear and avoidance of eye contact (*Schneier et al., 2011*; *Schulze, Renneberg & Lobmaier, 2013*). A study conducted in a virtual-reality environment showed that individuals with social anxiety disorder are more likely to be distressed when they have to perform a speech in front of an audience (*Cornwell et al., 2011*). Individuals with social anxiety disorder also showed a longer visual scan path and greater total fixation time at the non-social regions of the display in between and around the audience's faces during a speech (*Chen et al., 2015*). Another study (*Moukheiber et al., 2010*) showed that avoidant patterns of eye, such as the number of fixations and total fixation duration, decreased significantly for social phobic patients when they were presented with negative expressions (such as anger or disgust).

Considering the above studies, we hypothesized that social anxiety has an influence on the degree of discomfort with trypophobic images composed of clusters of human eyes and faces. The aim of the current study is to investigate three questions. The first question is whether images that contain clusters of human eyes and faces can induce aversion. The

second question is whether the TQ shows validity and reliability in other populations, as suggested by the previous study (*Le, Cole & Wilkins, 2015*). Third, we are interested in whether social anxiety directly involves discomfort due to the trypophobic images composed of clusters of human eyes and faces. For these purposes, we cropped the eye and face images separately. The relationships among the TQ, the Leibowitz Social Anxiety Scale (LSAS; *Liebowitz, 1987*), and the Discomfort Rating Score (DRS) were analyzed accordingly. In Experiment 1, we examined these questions with English and Japanese speakers. In Experiment 2, we tested whether a cluster of faces also induced discomfort in the same manner as cropped eyes.

# EXPERIMENT 1

## Method

### Participants

The experiment was conducted online using Qualtrics (Provo, UT: http://www.qualtrics.com). Participants whose native language was either English or Japanese were recruited from Qualtrics panels and received the designated compensation for each attempt. For those who made multiple attempts, only the first attempt was recorded. Eighty-seven English speakers took part in the study via the English section (66 males and 21 females; mean age = 18.8 yrs., SD = 11.1 yrs.). The Japanese-speaking group comprised one hundred and twenty-two individuals (71 males and 51 females; mean age = 25.3 yrs., SD = 9.6 yrs.). The purpose of the present study was not revealed to the participants. The experiments were conducted according to the principles of the Helsinki Declaration. The ethical committees of Kyushu University approved the protocol (approval number: 2013-008). Prior to the experiments, the participants consented to participation in the survey, and they could quit at any time if they felt sick due to the observation of disgusting images.

### Apparatus, stimuli, and procedure

The survey consists of three parts: the social anxiety scale, the Trypophobia scale, and the discomfort rating of eyes. The order of these three scales was randomized across all participants. There was no time limit to complete each scale.

Liebowitz social anxiety scale: We used the English version of the LSAS (*Liebowitz, 1987*) or the Japanese version of the scale (*Asakura et al., 2002*) according to the participants' language background. This scale consists of two sections. In the "fear or anxiety" section, the questions asked "how anxious or fearful do you feel in the specific situation." In this section, each item was scored on a 4-point Likert-scale: "None," "Mild," "Moderate" and "Severe." The second section was the "avoidance" section, in which the questions asked "how often do you avoid the situation." In this section, each item was scored on a 4-point Likert-scale: "Never (0%)," "Occasionally (1–33%)," "Often (33–67%)" and "Usually (67–100%)." The order of items was randomized across sections.

*Trypophobia scale.* We employed the original English version of the TQ, and the images of a lotus seed head and a honeycomb for all participants from the previous study (*Le, Cole & Wilkins, 2015*). Moreover, we developed a Japanese version of the TQ by directly

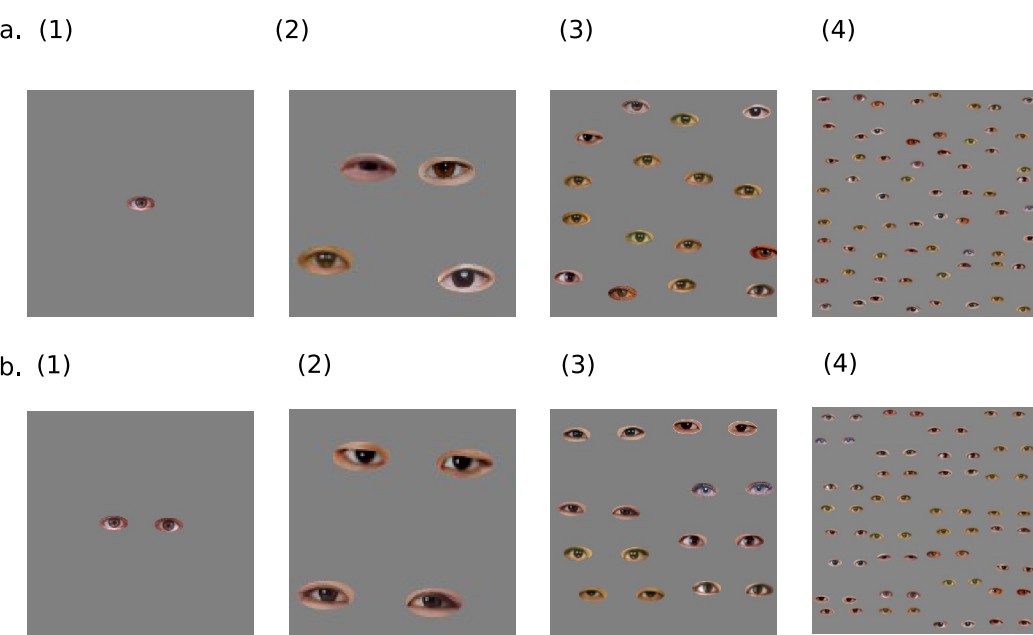

**Figure 1  The examples of stimuli used in Experiment 1.** The letter of each label indicates the Pair-condition (single or paired), and the number in each label indicates the Number-of-eyes-condition (1, 4, 16, or 64). The 256 × 256 pixels of the real eye image were created using the 64 × 64 pixels of one eye in the 1- and 16-eye-condition. The 4-eye-condition used the 128 × 128 pixels of the real eye image. The 64-eye condition used the 512 × 512 pixels of the real eye image.

translating items from the original version into Japanese, and we confirmed its literal comprehensiveness by back-translation with the original authors. The English and Japanese versions each contained 19 items; each item was rated based on a 10-point Likert-scale from 1 (strongly disagree) to 10 (strongly agree).

*Rating of eye cluster.*  We exported the real eyes stimuli from the database of ATR-Promotions (DB99; *ATR-Promotions, 2006*). We extracted only the eyes with their fringes and unified the size for each eye picture using GIMP2.8.14 (www.gimp.org). In the real eye block, there were two variables: Pair (single and paired) and Number of eyes (1, 4, 16, and 64; Fig. 1). Participants viewed each eye stimulus and evaluated it using a DRS from 0 (not at all uncomfortable) to 10 (extremely uncomfortable). The order of the stimulus presentation was randomized for each participant. Each eye stimulus was presented two times, so participants viewed 16 trials.

## Results
### The effect of eye types on the perceived discomfort
We performed analyses of variance (ANOVAs) to determine whether the number and pair of eyes affected the participants' DRS. A three-way within-participant ANOVA with language (English and Japanese), pair (single and paired), and number of eyes (4, 16, and 64) as factors was conducted. To discriminate clearly whether the number of eyes or the pair condition affected the DRS, the one- and two-eye conditions were excluded from the analysis to match the total number of eyes between the single and paired conditions.

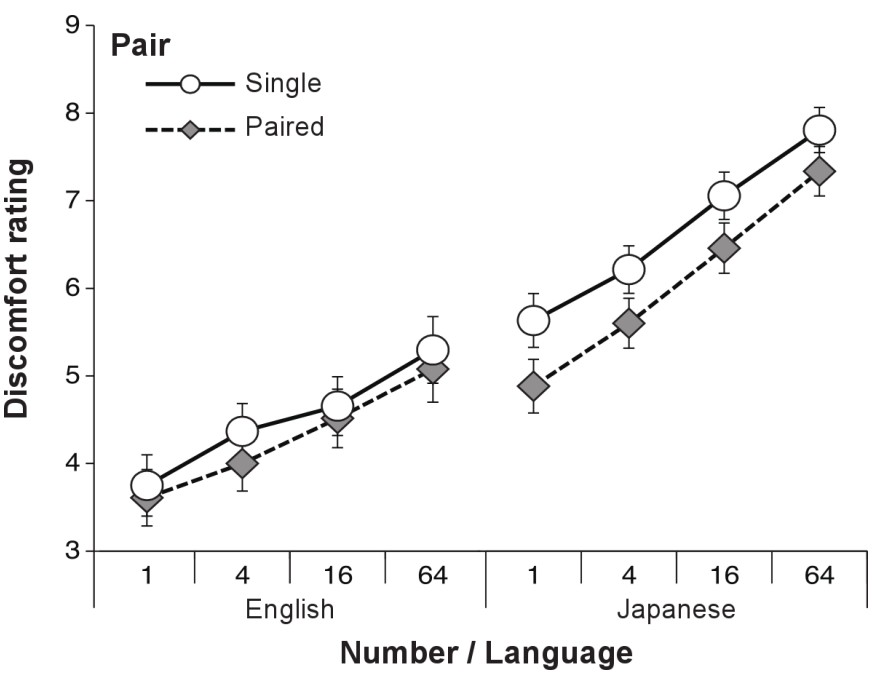

**Figure 2** The DRS for real eye in each condition in Experiment 1.

As shown in Fig. 2, the ANOVA revealed significant main effects of language ($F(1,206) = 41.16$, $p < .001$, $\eta_p^2 = .17$), pair ($F(1,206) = 25.57$, $p < .001$, $\eta_p^2 = .11$) and number of eyes ($F(2,412) = 151.86$, $p < .001$, $\eta_p^2 = .42$). Multiple comparisons using Ryan's method revealed that the 64-eye condition was significantly perceived to be more uncomfortable than the 4- and 16-eye conditions, and the 16-eye condition was significantly perceived to be more uncomfortable than the 4-eye condition. An interaction between language and number of eyes was significant ($F(2,412) = 4.32$, $p = .014$, $\eta_p^2 = .020$). Tests of simple main effect revealed that the DRSs of Japanese speakers were significantly higher than the DRSs of English speakers in all eye conditions (4-eyes: $F(1,618) = 29.58$, $p < .001$; 16-eyes: $F(1,618) = 39.24$, $p < .001$; 64-eyes: $F(1,618) = 46.46$, $p < .001$). For both language groups, the 64-eye condition was perceived to be significantly more uncomfortable than the 4- and 16-eye conditions, and the 16-eye condition was perceived to be significantly more uncomfortable than the 4-eye condition (English: $F(2,412) = 52.59$, $p < .001$; Japanese: $F(2,412) = 103.60$, $p < .001$). The other interactions were found to be not significant (language and pair: $F(1,206) = 2.63$, $p = .11$; pair and number of eyes: $F(2,412) = 1.65$, $p = .19$; language, pair and number of eyes: $F(2,412) = 1.23$, $p = .29$).

### Validity and reliability of TQ
To demonstrate the validity and reliability of the TQ, a confirmatory factor analysis (with promax rotation, Maximum-likelihood method) was individually conducted on the English and Japanese TQ scores. As in a previous study (*Le, Cole & Wilkins, 2015*), a scree plot showed that a one factor solution was sufficient to explain all variances in the analysis for both language versions of the TQ. The results are shown in Tables 1 and 2. The factor

**Table 1  Factor loadings for the items in the TQ on English after promax rotation in Experiment 1.**

| Item | Factor1 | Communality |
|---|---|---|
| Feel anxious, full of dread or fearful | .91 | .83 |
| Feel sick or nauseous | .89 | .79 |
| Feel like going crazy | .89 | .79 |
| Feel like panicking or screaming | .88 | .77 |
| Vomit | .87 | .76 |
| Have trouble breathing | .86 | .74 |
| Have goosebumps | .86 | .74 |
| Feel like crying | .83 | .70 |
| Have an urge to destroy the holes | .83 | .68 |
| Feel itchiness | .82 | .67 |
| Chills | .81 | .66 |
| Feel freaked out | .80 | .63 |
| Feel nervous (e.g., heart pounding, butterflies in stomach, sweating, stomach ache, etc.) | .79 | .63 |
| Shiver | .79 | .62 |
| Feel aversion, disgust or repulsion | .68 | .47 |
| Feel skin crawl | .65 | .42 |
| Feel uncomfortable or uneasy | .59 | .35 |
| Factor contribution | 11.23 | |
| Cumulative contribution | 68.60 | |
| $\alpha$ coefficient | 0.97 | |

loading value of each item, $\alpha$ coefficients, factor contribution and cumulative contribution are approximately consistent with those of *Le, Cole & Wilkins (2015)*. In the following analysis, we used a sum of the item scores on the TQ and the LSAS (*Liebowitz, 1987*).

### Correlation among DRS, TQ score and LSAS score

We calculated the DRS by summing the scores from each pair condition (single and paired) across all participants (i.e., data from both language groups were analyzed altogether). The correlations among the DRS (single and paired), the TQ score, and the LSAS score are shown in Table 3. There were significant correlations between the TQ score and the DRS in the single and paired conditions. Moreover, the correlations between the LSAS score and the DRS in both the single and paired conditions were also significant.

### Mediation effect of TQ

To investigate the direct influence of the LSAS score on the DRS, we conducted a mediation analysis. We set the LSAS and TQ scores as predictors of the two conditions (single & paired) used in the correlation analysis. The mediation model is shown in Fig. 3, and the results are shown in Table 4. In both the single and paired conditions, the path from the LSAS score to the DRS was significant (single: $\beta = .15$, $p = .034$; paired: $\beta = .19$, $p = .006$). When the TQ score was set as a mediator, the path from the LSAS score to the TQ score was significant (single: $\beta = .27$, $p.001$; paired: $\beta = .27$, $p < .001$) and the path from the TQ score to the

**Table 2  Factor loadings for the items in the TQ on Japanese after promax rotation in Experiment 1.**

| Item | Factor1 | Communality |
|---|---|---|
| Feel anxious, full of dread or fearful | .91 | .82 |
| Feel sick or nauseous | .89 | .80 |
| Feel nervous (e.g., heart pounding, butterflies in stomach, sweating, stomach ache, etc.) | .89 | .79 |
| Feel like panicking or screaming | .88 | .77 |
| Chills | .87 | .75 |
| Have trouble breathing | .87 | .75 |
| Have goosebumps | .86 | .74 |
| Feel like going crazy | .86 | .73 |
| Feel itchiness | .82 | .68 |
| Feel freaked out | .82 | .67 |
| Vomit | .81 | .66 |
| Feel uncomfortable or uneasy | .78 | .61 |
| Shiver | .77 | .59 |
| Feel skin crawl | .73 | .53 |
| Feel like crying | .72 | .52 |
| Feel aversion, disgust or repulsion | .71 | .51 |
| Have an urge to destroy the holes | .50 | .25 |
| Factor contribution | 11.16 | |
| Cumulative contribution | 67.58 | |
| $\alpha$ coefficient | 0.97 | |

**Table 3  Correlation among DRS, TQ and LSAS score in Experiment 1.**

| | 1 | 2 | 3 | 4 |
|---|---|---|---|---|
| 1. DRS for single eye | – | | | |
| 2. DRS for paired eyes | .90[**] | – | | |
| 3. LSAS score | .15[*] | .19[**] | – | |
| 4. TQ score | .39[**] | .44[**] | .27[**] | – |

**Notes.**
[**] $p < .01$.
[*] $p < .05$.

DRS was also significant (single: $\beta = .38$, $p < .001$; paired: $\beta = .41$, $p < .001$). However, the path from the LSAS score to the DRS was not significant (single: $\beta = .044$, $p = .51$; paired: $\beta = .078$, $p = .23$). A Sobel test revealed that the mediation effects were significant (single: $Z = 3.31$, $p = .001$; paired: $Z = 3.45$, $p = .001$).

## Discussion

In Experiment 1, we examined the relationship between social anxiety and the discomforting symptoms induced by images of eye clusters. In line with a previous study (*Le, Cole & Wilkins, 2015*), we found that not only clusters of holes but also the images that contain clusters of other objects induce aversive reactions. Cultural differences might be a factor

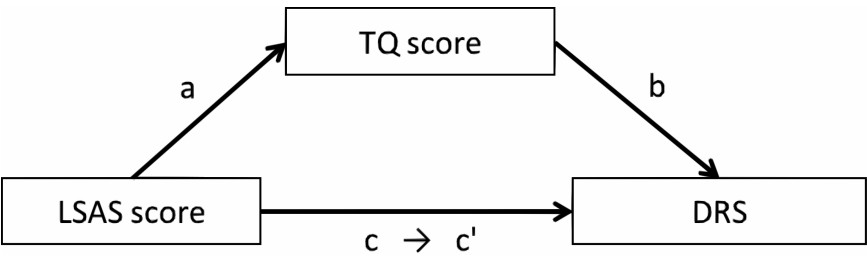

**Figure 3** The mediation model among the LSAS score, the TQ score, and the DRS in Experiments 1 and 2.

**Table 4** The result of mediation analysis in Experiment 1.

| DRS | Path | $\beta$ | SE | df | t |
|---|---|---|---|---|---|
| Single eye | a | .27 | .0933 | 206 | 4.09[**] |
| | b | .38 | .0050 | 205 | 5.63[**] |
| | c | .15 | .0073 | 206 | 2.13[*] |
| | c′ | .04 | .0071 | 205 | 0.66 |
| Paired eyes | a | .27 | .0933 | 206 | 4.09[**] |
| | b | .42 | .0048 | 205 | 6.40[**] |
| | c | .19 | .0070 | 206 | 2.80[*] |
| | c′ | .08 | .0067 | 205 | 1.20 |

**Notes.**
[**]$p < .01$,
[*]$p < .05$.

in the discrepancy of the aversive cognition because we found that Japanese speakers experienced more discomfort than English speakers.

The results from the factor analysis also supported the previous study (*Le, Cole & Wilkins, 2015*). The analysis showed that all items consisted of one common factor and satisfied the criterion for acceptable loadings for both the English and Japanese scales. In addition, the analysis indicates that direct translation of the TQ does not affect its validity or reliability when the TQ is applied to a study on another population.

We calculated the correlations among the DRS, the LSAS score, and the TQ score, from which we found a positive correlation ($r = .27$) between the LSAS and TQ scores. *Le, Cole & Wilkins (2015)* showed a small correlation ($r = .21$) between the TQ and the State-Trait Anxiety Inventory (STAI; *Spielberger, 1983*) and suggested that general anxiety accounts for trypophobia. Our results in Experiment 1 supported their findings.

The mediation analysis showed a significant mediation effect when the TQ score was a mediator; however, the path from the LSAS score to the DRS was not significant. These results were contrary to our hypothesis that there is positive correlation between social anxiety and the DRS concerning trypophobic images composed of eye clusters.

One of the reasons the significant path between social anxiety and discomfort was not observed may be attributed to the stimulus property. In the previous studies that investigated the relationship between social anxiety and gaze perception, stimulus eyes

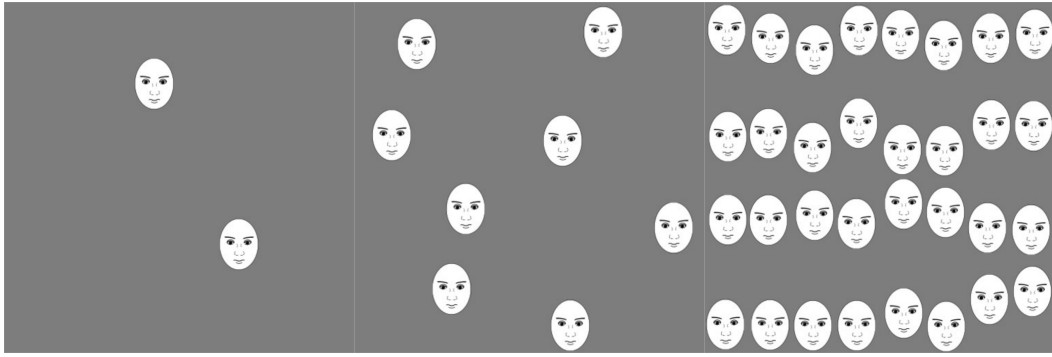

**Figure 4** **Examples of the stimuli used in Experiment 2.** (A) 2-face condition, (B) 8-face condition, and (C) 32-face condition. Each face image was scaled to an elliptical shape, 57 × 75 pixels. Realistic faces were used as stimuli in the actual experiment.

were usually embedded in a face (*Gamer et al., 2011*; *Schulze, Renneberg & Lobmaier, 2013*). Moreover, faces are processed holistically in specialized brain areas such as the fusiform face area (*Andrews et al., 2010*; *Arcurio, Gold & James, 2012*; *Schiltz & Rossion, 2006*). Thus, it is possible that if the individuals with a high social anxiety trait show a strong avoidance reaction to the gaze specifically when the eyes appear in a face, the direct effect from the LSAS score to the DRS may be weak because, in Experiment 1, the eyes were not in a face. Hence, in Experiment 2, we used the trypophobic images containing faces with eyes and investigated whether social anxiety could directly involve the aversion to images of gazing faces. If so, we could obtain a significant direct effect from the LSAS score to the DRS even after mediation by the TQ score.

# EXPERIMENT 2
## Method
### Participants
In this experiment, a new sample of 499 English speakers was recruited from Qualtrics panels (159 females and 340 males; mean age = 34.6 yrs., SD = 9.8 yrs.). The purpose of the present study was not revealed to the participants.

### Stimuli and procedure
The procedure was identical to Experiment 1, except that the rating of eye clusters was replaced by the rating of face clusters. As facial stimuli, we selected 32 neutral faces (half males, half females; AF01NES ~AF16NES, AM01NES ~AM11NES, AM13NES ~AM15NES, AM17NES, AM18NES) from the Karolinska Directed Emotion Face Set (*Lundqvist, Flykt & Öhman, 1998*). To exclude hair, we cropped these face stimuli to elliptical shapes and unified the size of each face picture.

In the rating of face clusters, the number of eyes was the same as in Experiment 1 (i.e., 4, 16 and 64; Fig. 4). The order of the stimuli presentation was randomized for each participant. Only English versions of the scales were used in this experiment because all the participants were English speakers.

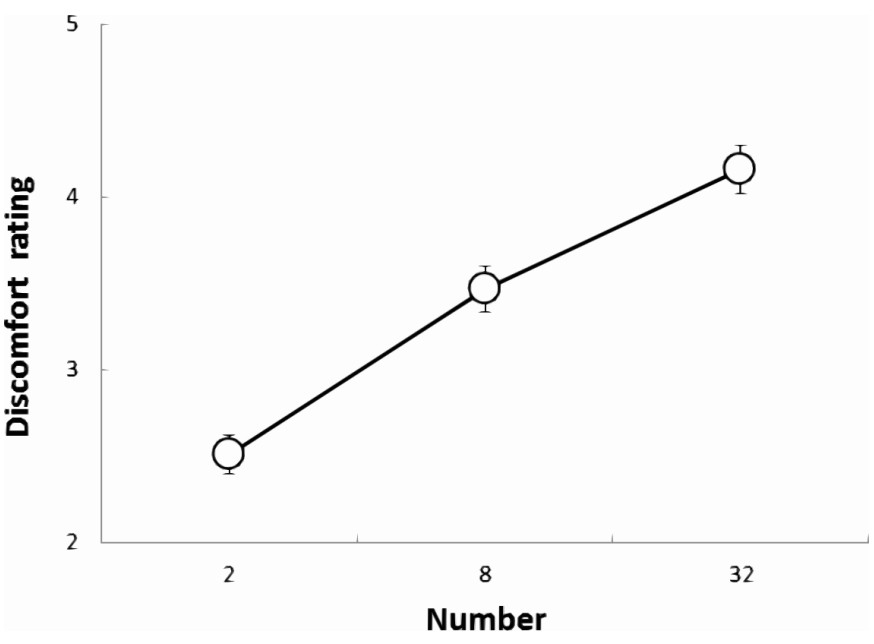

**Figure 5** The DRS for faces in each condition in Experiment 2.

## Results

### The effect of face conditions on perceived discomfort

As shown in Fig. 5, for the data in the cluster of faces, a one-way within-participant ANOVA with the numbers of faces (2, 8, and 32) as factors was conducted, which revealed significant main effects associated with the number of faces ($F(2, 996) = 165.87$, $p < .001$, $\eta_p^2 = .24$). The multiple comparisons revealed that the 32-face condition was significantly more uncomfortable than the 2- and 8-face conditions and that the 8-face condition was significantly more uncomfortable than the 2-face condition (all $ps < .001$).

### Validity and reliability of TQ

A confirmatory factor analysis (with promax rotation, Maximum-likelihood method) was conducted on the TQ scores. As shown in Table 5, all the items constituted one common factor. The factor loading values of each item, $\alpha$ coefficients, factor contribution and cumulative contribution were similar to those of both Le et al. and Experiment 1. In the following analysis, we used a sum of the item scores as the trypophobia scale score.

### Correlation among DRS, TQ and LSAS scale

We calculated the DRS by summing the scores of each of the faces for each participant. Table 6 shows the correlations of the DRSs and the TQ and LSAS scores. There were significant correlations among the DRSs and the TQ and LSAS scores.

### Mediation effect of TQ

In the mediation analysis, we entered the LSAS and TQ scores as predictors of the DRS of faces. The mediation model was set as in Experiment 1, and the results are shown in Table 7. The path from the LSAS score to the DRS was significant ($\beta = .36$, $p < .001$). When

**Table 5  Factor loadings for the items in the TQ after promax rotation in Experiment 2.**

| Item | Factor 1 | Communality |
|---|---|---|
| Feel sick or nauseous | .92 | .85 |
| Feel like panicking or screaming | .92 | .84 |
| Have trouble breathing | .92 | .84 |
| Feel nervous (e.g., heart pounding, butterflies in stomach, sweating, stomach ache, etc.) | .91 | .83 |
| Feel like going crazy | .91 | .82 |
| Feel anxious, full of dread or fearful | .90 | .82 |
| Feel freaked out | .90 | .81 |
| Chills | .90 | .81 |
| Have goosebumps | .90 | .80 |
| Shiver | .87 | .76 |
| Feel itchiness | .87 | .76 |
| Feel like crying | .87 | .75 |
| Have an urge to destroy the holes | .86 | .73 |
| Vomit | .85 | .73 |
| Feel uncomfortable or uneasy | .80 | .65 |
| Feel aversion, disgust or repulsion | .77 | .60 |
| Feel skin crawl | .77 | .59 |
| Factor contribution | 12.99 | |
| Cumulative contribution | 0.76 | |
| $\alpha$ coefficient | 0.98 | |

**Table 6  Correlation among DRS, TQ and LSAS score in Experiment 2.**

| | 1 | 2 | 3 |
|---|---|---|---|
| 1. DRS for faces | – | | |
| 2. LSAS score | .36** | – | |
| 3. TQ score | .56** | .32** | – |

Notes.
** $p < .01$.
* $p < .05$.

**Table 7  The result of mediation analysis in Experiment 2.**

| DRS | Path | $\beta$ | SE | df | t |
|---|---|---|---|---|---|
| Faces | a | .32 | .073 | 497 | 7.58** |
| | b | .42 | .008 | 496 | 10.47** |
| | c | .36 | .015 | 497 | 8.64** |
| | c′ | .23 | .014 | 496 | 5.66** |

Notes.
* $p < .01$.
** $p < .05$.

the TQ score was a mediator, the path from the LSAS score to the TQ score was significant ($\beta = .32$, $p < .001$) and the path from the TQ score to the DRS was also significant ($\beta = .42$, $p < .001$). The path from the LSAS score to the DRS was still significant, although the $\beta$ score decreased ($\beta = .23$, $p < .001$). A Sobel test revealed the significant mediation effect ($Z = 6.14$, $p < .001$).

## Discussion

We examined whether social anxiety directly linked to aversion to trypophobic images composed of faces in Experiment 2. The results revealed that the cluster of faces induced aversion and that the DRS increased when the number of faces increased; the validity and reliability of TQ was confirmed, as in Experiment 1. These results were in agreement with Experiment 1 and supported previous findings (*Le, Cole & Wilkins, 2015*).

In addition, the correlation analysis showed that the correlation coefficient between the LSAS score and the DRS was not higher than that between the TQ score and the DRS. The results suggested that even when the eyes were embedded in a face, social anxiety was not strongly related to discomfort for facial clusters, although the correlation was significant.

More importantly, not only the indirect effect but also the direct effect from the LSAS score to the DRS was significant in Experiment 2 when the TQ score was set as a mediator. This significant direct effect after mediation was not shown in Experiment 1, where we used only the cropped eye images. Based on these results, it is suggested that the perception of a face strengthens the induction of discomfort to gaze in individuals with social anxiety disorder and that trypophobia mediates this relation.

## General discussion

The present study aimed to investigate whether social anxiety is related to discomfort induced by clusters of eyes (Experiment 1) and faces (Experiment 2). The results suggested that both eyes and faces induced discomfort as the number of images increased. Additionally, this effect was strongly related to the social anxiety trait and was mediated by the trypophobia trait. Moreover, the social anxiety trait more strongly predicted discomfort for the clusters of eyes when the eyes appeared in a face.

Although our hypothesis that social anxiety is directly linked to discomfort due to the trypophobic images of human face clusters was supported, we did not expect the significant mediation effect in both experiments. It does not necessarily mean that only the TQ score mediated the path between the LSAS score and the DRS because general anxiety may have an influence on trypophobia. Nevertheless, we cannot completely exclude the possibility that the relationship between the LSAS score and DRS was mediated by the TQ score. For example, *Moukheiber et al. (2010)* showed that gaze avoidance and fear of blushing occurred with individuals who have a social anxiety disorder, which seems to support the hypothesis that social anxiety is a heterogeneous disorder. Further investigations are needed to clarify whether trypophobia can be classified as a subtype of social anxiety disorder based on our findings that the TQ score mediated the LSAS score and the DRS.

A limitation of the current study lies in the fact that the differences between individuals and cultures were not examined. For example, in Experiment 1, the results of the ANOVA

showed that Japanese speakers feel more discomfort than English speakers. However, the question of whether the difference in the DRS between Japanese and English speakers is caused by the aversion to trypophobic images or the sensitivity to gaze has not yet been determined. If the cultural difference in DRS was specific to the trypophobic images of eye clusters, one can argue that the difference came from cultural or ethnic differences in the sensitivity to gaze. This hypothesis predicts that Japanese speakers would show stronger social anxiety than English speakers, because individuals with social anxiety tend to attend to gaze (*Schneier et al., 2011*; *Schulze, Renneberg & Lobmaier, 2013*). But this was not the case. Experiment 1 showed that the LSAS score of the English speakers was significantly higher than that of the Japanese speakers (M = 61.63 vs. 54.46). Moreover, considering counter-evidence that social anxiety symptoms are more likely to be found in Japanese than American (e.g., *Dinnel, Kleinknecht & Tanaka-Matsumi, 2002*), it is bold to conclude that the DRS simply reflected the difference in the LSAS scores. Another possible explanation is that the spatial frequency information to evaluate discomfort depends on cultures. For example, previous studies revealed that social anxiety, culture, and emotion modulated the mental and neural processing of spatial frequency information or facial recognition (*Curby, Johnson & Tyson, 2012*; *Miellet et al., 2013*; *Riwkes, Goldstein & Gilboa-Schechtman, 2015*; *Wieser & Moscovitch, 2015*). Based on the fact that the esthetic pleasure was affected by spatial frequencies in the Italian and Japanese observers (*Vannucci, Gori & Kojima, 2014*), it is premature to conclude that such influence of spatial frequency is specific to face. Possibly, the range of spatial frequency information to induce discomfort may be a common between individuals with social anxiety disorder and trypophobia. Therefore, the relationship between trypophobia and cultural differences or individual differences in anxiety should be further investigated on the perspective of spatial characteristic of visual stimuli in the future.

In conclusion, the current study revealed that not only clusters of holes but also clusters of other objects such as eyes and faces induce aversive reactions. The validity and reliability of the TQ and the fact that social anxiety directly involved the aversion to trypophobic images of faces were also established. Further studies are required to explore other factors that influence trypophobia. Such studies could contribute to understanding why some individuals develop an aversion to trypophobic images but others do not.

## ACKNOWLEDGEMENTS

The authors would like to thank Drs. Keiko Ihaya and Kyoshiro Sasaki for their solid supports for developing the Japanese version of the Trypophobia Questionnaire.

### Funding

This work was supported by Kyushu University Interdisciplinary Programs in Education and Projects in Research Development (27822), JSPS KAKENHI (26540067), and JSPS KAKENHI (15H05709). The funders had no role in study design, data collection and analysis, decision to publish, or preparation of the manuscript.

### Grant Disclosures

The following grant information was disclosed by the authors:

Kyushu University Interdisciplinary Programs in Education and Projects in Research Development: 27822.

JSPS KAKENHI: 26540067.

JSPS KAKENHI: 15H05709.

### Competing Interests

The authors declare there are no competing interests.

### Author Contributions

- Kengo Chaya wrote the paper, reviewed drafts of the paper.
- Yuting Xue conceived and designed the experiments, performed the experiments, contributed reagents/materials/analysis tools, wrote the paper.
- Yusuke Uto conceived and designed the experiments, wrote the paper.
- Qirui Yao performed the experiments, contributed reagents/materials/analysis tools, wrote the paper, reviewed drafts of the paper.
- Yuki Yamada performed the experiments, analyzed the data, prepared figures and/or tables, reviewed drafts of the paper.

### Human Ethics

The following information was supplied relating to ethical approvals (i.e., approving body and any reference numbers):

The ethical committees of Kyushu University approved the protocol (approval number: 2013-008).

### Data Availability

The raw data has been supplied as Data S1 and Data S2.

### Supplemental Information

Supplemental information for this article can be found online at http://dx.doi.org/10.7717/peerj.1942#supplemental-information.

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
