# Peer review of "Fear of eyes: triadic relation among social anxiety, trypophobia, and discomfort for eye cluster"

_PeerJ, doi:10.7717/peerj.1942_

## Round 0.1 · original submission · Major Revisions

· Academic Editor

Major Revisions

Thank you for your submission to PeerJ. In my opinion as the Academic Editor for your article, your MS requires a number of major revisions before we could accept it for publication. I will be happy to consider a substantially revised MS that addresses all the points raised by the reviewers.

·

Basic reporting

ok

Experimental design

ok but see below

Validity of the findings

The authors present a case that some social anxiety when addressing a crowd may be due in part to trypophobia, a phobia from clusters of objects, in this case, eyes. They show that the trypophobia questionnaire of Le et al provides a better measure of discomfort from images of clusters of eyes than does a social anxiety scale. The findings are interesting, and I would like to encourage the authors to undertake a second study to clarify some of the issues that their findings raise. First, it is rare that crowds of people such as might induce social anxiety have the appearance of clusters of eyes. The eyes appear in faces and faces are processed holistically in specialized brain areas. Images of crowds with faces, directed towards a stage on which the camera is sited, would provide an interesting comparison. Provided the eyes were similarly prominent but surrounded by the face from which they came would they be trypophobic? For these images would the social anxiety scale provide a better correlate of discomfort than the trypophobia questionnaire? This questions are posed by the present study and require an answer prior to publication, in my view.

Additional comments

see above

Reviewer 2 ·

Basic reporting

No Comments

Experimental design

No Comments

Validity of the findings

No Comments

Additional comments

The manuscript entitled “Fear of eyes: The influence of social anxiety on trypophobic eyes” is presenting an experiment which is correctly carried out and the analysis are solid. English, however, need to be checked, it is not that the writing is bad but a check by a native English speaker seems important to make the reading of the manuscript easier and more fluent. I have some suggestions that I hope that will help in improving the manuscript:
1) When the effect of spatial frequency on trypophobia is described some words should be spent in showing how spatial frequency is crucial for our visual system. It could be written that our visual system is often described as a ‘‘spatial frequency analyzer” (Maffei & Fiorentini, 1973), the spatial frequencies influence the perception in many domains such as visual illusions (Giora & Gori, 2010) or even esthetic pleasure (Vannucci et al. 2014). Also fixational eye movements, necessary for avoiding fading (e.g. Martinez-Conde et al. 2006; 2013; Costela et al. 2013; McCamy et al. 2014) serve also in increasing specific spatial frequency detectability (Rucci et al. 2007). All this literature should be included in the introduction.
2) The link between highly poisonous animals and the spectral feature of the trypophobic images should be explained in more details because it is quite interesting.
3) If I correctly understood, the first point of the hypothesis: “First, whether the images which contain cluster of eyes can induce aversion as same as clusters of holes do” is not tested in the study because there is not a direct comparison between images with holes and images with eyes in the same participants (or even in two different groups) then this sentence should be deleted.
4) In the methods it is not clear how the Authors unified the luminance considering that it seems not to be measured by a photometer (at least is not reported) and moreover being an online survey the luminance of the stimuli may vary, dramatically among screens. It is not a crucial point but it should be clarified better in order to avoid confusion in the readers.
5) I don’t understand the meaning of this sentence: “The one- and two-eye(s) conditions were excluded from the analysis to match the total number of eye between single and paired conditions.” I understand that 1 and 2 eyes conditions were dropped but not why.

---

## Round 0.2 · Minor Revisions

· Academic Editor

Minor Revisions

Both reviewers are almost completely satisfied with the MS in present form, but please address Reviewer 1's concerns about correlation versus causation in the re-submission.

·

Basic reporting

The report is generally well-written, but I have annotated the draft with small suggestions for improvements, largely to the English.

Experimental design

I am not a fan of path analysis, but this seems OK

Validity of the findings

The study is now very interesting, and will doubtless provoke further study.

Additional comments

I am delighted the authors chose to conduct a further study. Their results are now consistent and convergent and provide a very interesting set of data. I have only one major quibble and that is that they refer to causation, when no causation has been demonstrated. To quote the adage, causation cannot be inferred from correlation. Please could they remove any reference to causation (unless they wish to qualify their use of this term).

Reviewer 2 ·

Basic reporting

The manuscript is vastly improved due to a great work done by the Authors, in my humble opinioni the manuscript is ready to be published

Experimental design

Fine

Validity of the findings

Fine

Additional comments

The Authors did a great job I like this manuscript and I hope to see it published soon.

---

## Round 0.3 · accepted · Accept

· Academic Editor

Accept

Please address the remaining clarification request (Reviewer 1) in your final production proof.

·

Basic reporting

The article is ready for publication, in my opinion.
I would suggest that the authors reword the phrase
"Another study (Moukheiber et al., 2010) showed that avoidant patterns of eye, such as the number of fixations and total fixation duration". It is not clear what "avoidant patterns of the eyes" are.
The acronym DRS is used extensively, and it is not clear what it refers to without reference to the introduction. Perhaps "rating scale" could be used instead or "Discomfort Rating Scale" could be spelled out at the beginning of each section?

Experimental design

OK

Validity of the findings

OK